# Characterization of *Lophiotoma leucotropis* Mitochondrial Genome of Family Turridae and Phylogenetic Considerations within the Neogastropoda

**DOI:** 10.3390/ani14020192

**Published:** 2024-01-07

**Authors:** Xinqin Jiang, Jing Miao, Jiji Li, Yingying Ye

**Affiliations:** 1National Engineering Research Center for Marine Aquaculture, Zhejiang Ocean University, Zhoushan 316022, China; jiangxq@zjou.deu.cn (X.J.); miaojing@zjou.edu.cn (J.M.); lijiji@zjou.edu.cn (J.L.); 2College of Fisheries, Zhejiang Ocean University, Zhoushan 316022, China

**Keywords:** Turridae, Neogastropoda, mitogenome, phylogeny

## Abstract

**Simple Summary:**

*Lophiotoma leucotropis* belongs to the Turridae and is a carnivorous snail that is widely distributed in shallow sea sand and intertidal zones around the world. Currently, the mitochondrial genome sequence of *L. leucotropis* has not been explored. And there is also great controversy over whether the order Neogastropoda to which *L. leucotropis* belongs is monophyletic. We sequenced the mitogenome of *L. leucotropis* and analyzed its basic gene characteristics. We found that it contains a total of 37 gene contents. Similar to species in the Turridae family, it has a higher content of AT bases and a preference for some codons. The selection pressure analysis results showed purification selection effects, and through comparison with other species in Turridae, it was found that the gene sequence of this species is conserved, which helps to deepen the understanding of the Turridae. We constructed a phylogenetic tree using the 13 protein coding genes of Neogastropoda and concluded that the order is monophyletic. This result is a supplement and validation to the research on the evolutionary status of Neogastropoda and also provides reference for the species classification and germplasm resource development of Neogastropoda.

**Abstract:**

Neogastropoda is a group of marine organisms with an extremely wide distribution that is rich in species and economic and ornamental values, the classification of species in this order has been ongoing for a long time, but there is still a great controversy about whether this order is monophyletic. In this study, we obtained the complete mitogenome of *Lophiotoma leucotropis* by next-generation sequencing and analyzed the basic structural features of the genome, and we found that the number of genes was consistent with that of most of the Neogastropoda snails, containing 37 genes, including 13 protein-coding genes (PCGs), 2 rRNAs, and 22 tRNAs. Analyzing base content, amino acid content, codon usage preference, and tRNA structure, the mitogenomes of eight species of Turridae were selected for analysis of selection pressures, and it was found that the evolution of species in this family was affected by purifying selection. In addition, by analyzing the rearrangement characteristics, it was found that the sequence of *L. leucotropis* was consistent with the Conoidea consensus order, and four of the eight species involved in the analysis showed rearrangements. Finally, we constructed a phylogenetic tree by combining PCGs of 60 species within Caenogastropoda and found Neogastropoda to be a monophyletic group, validating the results of morphological classification. The results will provide more references for the classification and species evolution of Neogastropoda, as well as phylogenetic analysis.

## 1. Introduction

*Lophiotoma leucotropis* (Neogastropoda, Conoidea, Turridae) was originally named *Pleurotoma leucotropis* [1]. The shell of *L. leucotropis* is tall and pointed, with a long water pipe groove, a brown shell surface, a white line surrounding each shoulder, and a gap in the water pipe groove. This species often inhabits the sandy seabed with water depths of 20–100 m and is a common benthic organism in coastal areas of China. Like most neogastropod snails, it has significant predatory behavior [2,3]. Taylor, in 1980, studied the feeding of predatory gastropods in Hong Kong waters and found that *L. leucotropis* mainly preyed on a species of *Poecilochaetus tropicus* in the family Poecilochaetidae, with a lower selectivity for *Laocince* and Capitellid [4,5].

Turridae, one of the major mollusc groups in the marine mollusc fauna, is the most diverse of all the mollusc families, with more than 5000 predicted species [6,7]. Turridae have an important role in the food chain of ecosystems, especially benthic organisms, as important predators and prey, and more than 50% of the predatory gastropods in the deep seabed are Turridae species [8]. They are therefore an important component of the benthic community from shallow subtidal to deep-sea areas. Their distribution is global and they can inhabit any part of the world’s oceans, the most common distribution is in tropical, subtropical, and temperate waters. The flesh of the Turridae is edible, and many species are tasty and of economic value, and the unique shape and color of the shells of many species are of great ornamental and collector value. There is a great deal of individual variation in the family, with shell heights ranging from a few millimeters to more than 15 centimeters. The shell of the Turridae snail is very diverse, and species can be difficult to distinguish because of the great variation in shell shape, size, sculptural features, and shell color.

The traditional morphological distinction between the species of Turridae is mainly in recognizing differences in the shell sculpture, Protoconch or apex, radula, outer lip and the labial sinus, color, operculum, etc. The embryonic shells of the Turridae are divided into two types: paucispiral and polygyrate, the radula is generally divided into four types: A, primitive; B, lacking lateral radula; C, ciliated only; D, without ciliated radula or without lateral radula, and the keratinized operculum generally has a dendritic to subelliptical and an ovoid to subovoid form [9]. Casey [10] initially divided the Turridae into eight tribes based on operculum and shell characteristics, Powell [11] divided the Turridae into nine subfamilies with reference to the operculum and radula, and then McLean and James [12] added six subfamilies by identifying the radula, bringing the total number of subfamilies to fifteen. In a later study, Taylor [13] almost overturned the previous classification, based on morphological classification and molecular techniques, and proposed a new taxonomic system by constructing a phylogenetic tree for the Turridae, which was divided into five subfamilies: Clavatulinae, Crassispirinae, Zonulispirinae, Cochespirinae, and Turponidae, and transferred some subfamilies under Conidae; however, the classification of subfamilies has not continued. According to the latest revision of the WoRMS (World Register of Marine Species: https://www.marinespecies.org/, accessed on 15 August 2023) in 2010, the classification of subfamilies no longer exists and they can be directly classified based on genera. Therefore, the existing Turridae are divided into 32 genera.

In mentioning the Turridae, we should also consider the Neogastropoda. Neogastropoda snails are also called Stenoglossa. According to the WoRMS, they currently comprise seven superfamilies: Buccinoidea, Conoidea, Mitroidea, Muricoidea, Olivoidea Turbinelloidea, and Volutoidea, with a great diversity of about 16,000 species [3,14]. Since the Cretaceous period, this order of snails has penetrated almost all oceans of the world [15,16]. They can be found in the Bohai Sea, Yellow Sea, East China Sea, and South China Sea, often inhabiting rocky areas in the low tidal zone to temperate subtidal zones. They are also found in muddy sea areas or coral reefs in the subtidal zone. They are also found near rock cracks and tide pools. They are a group of marine carnivorous invertebrates with extremely rich diversity [3,17]. The classification of Neogastropoda has gone through quite a long period of change due to the great variety of species. The species identification and phylogeny of neogastropod snails are highly controversial. Initially, the morphological classification of Neogastropoda was considered to be monophyletic [3,18,19], but in molecular analyses it has been found many times that the results do not support monophyly. Phylogenetic analysis by Colgan [20] using partial *28S rDNA* and *histone H3* sequences revealed a controversial classification of Neogastropoda. Cunha et al. [21] found in a phylogenetic relationship analysis of 13 protein-coding genes (PCGs) based on Neogastropoda mitogenome that species of Littorinimorpha would be interspersed in Neogastropoda. In recent years, some phylogenetic analyses of Caenogastropoda have also repeatedly involved Gastropoda, and we found that in Caenogastropoda there is indeed an interspersed distribution of the different families of Neogastropoda and Littorinimorpha [22,23].

Compared with traditional morphological classification, mitogenomes can be used as an aid to identify species more accurately because of their fast evolutionary rate, maternal inheritance, and easy purification and separation [24]. Therefore, analyzing the mitogenome of *L. leucotropis* from a molecular perspective will help us better understand the mitogenome characteristics of Turridae species and help us compare the gene arrangement and composition differences among species in Turridae. At the same time, the differences and similarities of gene quantity and arrangement between Turridae and other gastropods can also be explored. At the same time, mitogenomes are often used to reconstruct developmental relationships and deduce evolutionary relationships. There is still controversy over whether Neoastropoda is monophyletic. Considering that the use of different genes for analysis may lead to changes in the branches of the phylogenetic tree, we believe that the selection of mitochondrial whole genome sequences, especially the use of the combination of 13 PCGs, is more conducive to obtaining objective phylogenetic analysis results. Therefore, the objectives of this study were to sequence the complete mitogenome of *L. leucotropis* and analyze its basic gene composition, base content differences, codon usage, selection pressure analysis, and comparison of gene rearrangements among species by constructing a phylogenetic tree to investigate the interspecific evolutionary patterns, to clarify the evolutionary status of the three orders within Caenogastropoda, and to investigate whether Neogastropoda is monophyletic. It is hoped that the results will provide a reference for the species classification of Neogastropoda, as well as for the evolution and phylogeny of Caenogastropoda and the development of germplasm resources.

## 2. Materials and Methods

### 2.1. Sample Preparation and DNA Extraction

A single specimen of *L. leucotropis* was collected from Zhoushan, Zhejiang, China, (30°19′ N, 122°72′ E) in November 2018. The specimen was compared with the published reference book “*Atlas of the Famous Shell in the World for Identification and Appreciation*” [25], and morphologists from the Museum of Marine Biology, Zhejiang Ocean University were consulted, with a final check of the biological Latin name with the WoRMS. Fresh tissue was stripped from the shell, and the muscular tissue was stored in anhydrous ethanol at −20 °C after removing the digestive glands. Total DNA was extracted by salting-out [26]. The DNA quality was determined by 1% agarose gel electrophoresis and stored at −20 °C for sequencing.

### 2.2. DNA Sequencing and Assembly

The DNA of *L. leucotropis* was submitted to Origingene Bio-pharm Technology Co., Ltd. (Shanghai, China), and the genomic DNA of the qualified samples was broken into 300–500 bp fragments by ultrasonication and the broken DNA was purified to construct a sequencing library. The steps were: DNA end repair, 3′ end A base addition, sequencing junction ligation, and agarose gel electrophoresis to recover the target fragments. The target fragments were amplified by PCR to finalize the sequencing library construction, and the library was quality checked. The mitogenome of *L. leucotropis* was sequenced using Illumina Novaseq 6000 platform (Illumina, San Diego, CA, USA). The average insert size of a sequencing library is about 400 bp, and each library generates about 10 Gb of raw data. The raw data are filtered for sequencing junction sequences, low-quality reads, high-N-rate sequences, and short-length sequences to obtain high-quality sequencing data. We utilized the MEGA X software to conduct a comparative analysis between the *COI* gene obtained from our sequencing efforts and the *COI* gene fragment of *L. leucotropis* available on NCBI (GenBank Accession number: KM218750) in order to reaffirm the species identification [27].

### 2.3. Genome Annotation and Bioinformatics Analysis

Genome annotation was performed using the online tool Mitos2 (http://mitos2.bioinf.uni-leipzig.de/index.py, accessed on 10 September 2023) [28], and invertebrate codon lists were selected. Complete mitogenome data were verified and corrected using the Sequin 13.7, and then we uploaded the mitogenome data to the NCBI database (https://www.ncbi.nlm.nih.gov/, accessed on 10 September 2023) to obtain the GenBank Accession number. Mitogenome circular maps were generated by the online website Proksee [29]. DAMBE 7 [30] was used to count the complete mitogenomic and 13 PCGs in ATCG bases. Skew values were calculated as AT-skew = (A − T)/(A + T) and GC-skew = (G − C)/(G + C), respectively [31]. We used MEGA X [27] to calculate the content of various amino acids in PCGs and the RSCU (relative synonymous codon usage) values in PCGs. Mitos2 [28] was used to obtain the secondary structure diagram of all tRNA. Ka/Ks (non-synonymous vs. synonymous substitution) ratios were calculated using DnaSP 6 [32]. Selection pressure in Turridae was analyzed using the Ka/Ks ratio, and a positive selection effect was considered to be present if Ka/Ks > 1. Neutral selection was considered to be present if Ka/Ks = 1, and purifying selection effects were considered to be present if Ka/Ks < 1.

### 2.4. Phylogenetic Analysis

We constructed a phylogenetic tree based on 13 PCGs of Caenogastropoda mitogenome sequences, which covered 60 species in three orders, Littorinimorpha, Neogastropoda and Architaenioglossa, and the sequences of all the species except the newly sequenced species in this study were obtained from the NCBI Genbank database (Table 1). And two gastropod species, *Patelloida ryukyuensis* and *Lottia luchuana*, were selected as the outgroups, whose Genbank Accession numbers were MZ329339 and MZ329341, respectively. PCGs for each sequence were identified using DAMBE 7 [30] and PCGs for all species were concatenated into one large dataset. Sequences were compared and trimmed using MEGA-X [27], and nucleotide substitution saturation was calculated using DAMBE 7 software [30] to assess the suitability of these sequences for the construction of phylogenetic trees. Phylogenetic analyses were based on the Bayesian Inference (BI) method of the MrBayes 3.2.7a [33] program and the Maximum Likelihood (ML) method of the IQ-tree 2.1.3 [34] program. The best-fit model (GTR + F+R8) selected according to the BIC criteria by ModelFinder [35] was used for ML analysis. In the ultra-fast likelihood bootstrap, 1000 bootstrap repetitions were used to reconstruct the consensus tree. The BI method first required format conversion by PAUP 4 [36], and MrMTgui was then used to select the best alternative model by combining the results of PAUP 4, Modeltest 3.7 [37], and MrModeltest 2.3 [38], and then MrBayes settings for the best-fit model (GTR + I+G) were selected by AIC in MrModeltest 2.3 [38]. The BI analysis consisted of two Markov Chain Monte Carlo (MCMC) runs of 2,000,000 generations, sampled every 1000 generations, with 25% of the aged samples discarded. At the end of the run, the evolutionary tree was edited and beautified using Figtree v1.4.3 [39].

## 3. Results

### 3.1. Characterization of the Mitogenome Structure

The complete mitogenome sequence of *L. leucotropis* has been uploaded to NCBI with the GenBank Accession number OR607650. Its structure is shown as a typical closed-ring double-stranded molecule (Figure 1), with a length of 16,362 bp. The genome contains a total of 37 genes, with 13 PCGs, 22 tRNAs, and 2 rRNAs (*12S rRNA* and *16S rRNA*). There are 29 genes located in the heavy chain and 7 genes in the light chain, and all of them are tRNA genes. The shortest of the 13 PCGs is *atp8*, which is 159 bp in length and encodes 53 amino acids, and the longest is *nad5*, which is 1689 bp in length and encodes 563 amino acids (Table 2). The length of the tRNA is between 66 and 70 bp, the length of the *12S rRNA* is 883 bp, and the length of the *16S rRNA* is 1360 bp, all of which are within the normal range. The overlapping regions are generally small, with the largest being only 10 bp between *trnV* and *16S rRNA*, and there is a maximum intergenic nucleotides region of 48 bp between *trnS2* and *nad2*. In addition, there is a D-loop of 1172 bp between *trnF* and *cox3*. In all 13 PCGs, ATG and ATT were used as start codons, and TAA and TAG were used as stop codons.

### 3.2. Base Content and Base Offset

Calculation of the nucleotide composition of the mitogenome of *L. leucotropis* revealed that the contents of the four nucleotides were: A 30.93%, T 38.27%, G 15.73%, and C 15.07% (Table 3), with contents of A + T of 69.20% and contents of G + C of only 30.80%, and the sums of the AT bases and the GC bases were significantly different. The value of AT-Skew was −0.11, showing a slight T-offset, and GC-Skew was 0.02, showing a slight G-offset. The base analysis of 13 PCGs showed that the content of base A ranged from 24.03% to 32.70%, base T ranged from 36.70% to 45.74%, base G ranged from 11.89% to 21.09%, base C ranged from 10.69% to 19.57%, and A + T ranged from 64.96% to 75.47%. It can be seen that the AT bases were significantly more than the GC bases. The AT-skew values ranged from −0.31 to −0.10, showing a slight T offset, and the GC-skew values ranged from −0.21 to 0.18 and did not exhibit significant shifts to G or C bases.

### 3.3. Amino Acid Content, RSCU, and tRNA

The analysis of amino acid content showed that the contents of Phe, Ser2, Leu1, Leu2, Asn, Tyr, and Ile were higher than 6%; especially, Phe was the highest at 10.13% (Figure 2), and the contents of Arg and Asp were lower at 1.57% and 1.94%, respectively. The statistics of the RSCU values for all amino acid codons showed that the four codons used more frequently were GCU (Ala), UUA (Leu2), CCU (Pro), and GUU (Val), and the codons used less frequently were GCG (Ala), CCG (Pro), UCG (Ser1), and ACG (Thr). The mitogenome of *L. leucotropis* had a total of 22 tRNAs, and we found that almost all of the tRNAs have a typical cloverleaf structure (Figure 3), and a few of the tRNAs appear to have smaller loops on the amino acid acceptor arms (*trnC*, *trnD*, *trnF*, *trnQ*).

### 3.4. Selective Pressure Analysis

In the selection pressure analysis, substitutions that do not result in amino acid sequence changes are called non-synonymous substitutions, denoted by Ka, and vice versa for synonymous substitutions, denoted by Ks. The selection pressure analysis was based on the PCGs of Turridae species available in NCBI, plus a total of eight species of *L. leucotropis* sequenced in this study. The analysis showed that the Ka/Ks values of the 13 PCGs were all significantly less than 1 (Figure 4), with *nad6* having the maximum value of 0.246 and *cox1* having the minimum value of 0.042. The mutations produced synonymous substitutions, and the Turridae species were affected by purifying selection during the evolutionary process.

### 3.5. Gene Rearrangement

Using *cox1* as a starting point, we made a mitochondrial gene sequence map of the eight known species of Turridae and compared them with the Conoidea consensus order (Figure 5). We found that the sequences of the four species of *Cochlespira* sp., *L. cerithiformis*, *L. leucotropis*, and *Otitoma* sp. are consistent with the Conoidea consensus order, all possessing 37 mitochondrial genes, with a slight change in the gene sequence of *Gemmuloborsonia moosai*, with *trnF* moving before *trnT*. In *Fusiturris similis*, the gene changes were more pronounced: *trnS* moved in front of *cob*, which can be interpreted as an exchange of the positions of the two genes, and *trnN*-*trnI*-*nad3*, which used to be located after *trnR*, was also shifted, moving in front of *trnK* and becoming *trnI*-*nad3*-*trnN*. *Cochlespira* sp. and *Pinguigemmula* sp. were more obviously both missing *trnF*, and the *trnN*-*trnI*-*nad3* fragment in *Cochlespira* sp. underwent the same change as in *F. similis*.

### 3.6. Phylogenetic Relationship

Phylogenetic analyses of 13 PCGs sequences of 60 species from three orders (Littorinimorpha, Neogastropoda, Architaenioglossa) within Caenogastropoda were carried out, and the two methods of ML and BI, used for this analysis, showed almost the same topology (Figure 6). Eventually, we used the topology inferred by the BI method and labelled the nodes with the ML method. Based on the topology, it can be seen that the three orders show the relationship of ((Neogastropoda) + Littorinimorpha) + Architaenioglossa, and each order forms a monophyletic branch, with most of them having a posterior probability value of 1 and a Bootstrap value of 100. Conoidea is the innermost superfamily distributed in Neogastropoda. In Conoidea, the affinities of the families are shown to be (Turridae + Clavatulidae) + Conidae. The seven genera of Turridae form three branches: *Lophiotoma*, *Pinguigemmula* and *Gemmuloborsonia* are clustered into a single unit, and *Otitoma*, *Leucosyrinx*, and *Fusiturris* are clustered into a single unit, this two branches are mutually sisters groups. Then, these two branches are subsequently clustered with *Cochlespira*, all eight species form a cluster.

## 4. Discussion

### 4.1. Basic Features of the Mitogenome of L. leucotropis

In this study, we sequenced the complete mitogenome of *L. leucotropis*, analyzed the basic compositional features of the genes, the base content and base preference of each gene, the codon usage preference, and the amino acid content, and carried out the calculation of selective pressure values, as well as a comparison of gene rearrangements among the species within the Turridae. The mitogenome of *L. leucotropis,* like most gastropods, contains 37 genes, which follow the composition of 13 PCGs, 22 tRNA, and 2 rRNA. The genome sequence length is 16,362 bp, and the sequence lengths of seven other species of the same family currently available in the NCBI database range from 15,097 to 15,595 bp. In *L. leucotropis*, there is a D-loop region between *trnF* and *cox3*, with a length of 1172 bp, which is significantly longer than several other species, making the length of the complete mitogenome of *L. leucotropis* relatively longer. The mitogenome base content pattern of *L. leucotropis* is also consistent with that of most gastropods, with a somewhat higher content of AT bases and the use of ATN as the initiation codon and TAN as the termination codon. The tRNA secondary structure of *L. leucotropis* is very much in line with the tRNA structural features, but in its very closely related congener, *L. cerithiformis*, we found interesting structures where the DHU-loop and the central loop of the two Ser are fused together; this phenomenon is normal in invertebrates [40]. 

In the selection pressure analysis, the overall Ka/Ks levels of the mitogenomes in this calculation were all less than 1, indicating that the mutations produced synonymous substitutions and that Turridae species were affected by purifying selection during their evolutionary course. However, there are only eight sequences known to us that can be used in the selection pressure analysis, which is a relatively small amount of data, and it would be helpful to obtain more objective results if we acquire more information about the sequence of the species in future sequencing. In looking at gene rearrangements, we found that four of the eight currently known species sequences are consistent with the Conoidea consensus order, with *trnF* deletions or gene translocations in the other four species. Rearrangements of genes in gastropods are not uncommon as the mitogenome sequences of most species of gastropods contain 37 genes; 22 of them are tRNA, but there are cases where tRNA duplication leads to an increase in the number of tRNA, for example, in Patellogastropoda, where duplication of *trnM* or *trnW* was found; thus, 23 or 24 tRNA may be present [41,42]. Increased tRNA numbers are especially common in bivalves and can even exceed 40 [43,44].

### 4.2. Phylogenetic Analysis

This group of Neogastropoda has relatively obvious common morphological characters, and in recent years there have been many reports on the morphological characteristics (e.g., radula), distribution, etc., of new species of neogastropods [45,46,47]. Regarding the Turridae involved in this study, despite the richness of the family, research has mainly focused on its morphological classification [9] and few complete mitochondrial sequence data are known for Turridae. From our present analysis, which classifies the species as occupying seven genera, at least we can see that species of the same genus are clustered together and that the species of the family as a whole form a monophyletic branch containing three sister groups, which provides a reference for future studies.

It is the classification of the larger range of Neogastropoda, as compared to Turridae, that is most controversial at present. Many molecular studies do not support the monophyly of Neogastropoda. Riedel’s [48] analysis of Neogastropoda and Caenogastropoda using *16S rRNA* and *18S rRNA* failed to recover the monophyly. Colgan et al. [49,50] conducted two successive phylogenetic analyses of Caenogastropoda using *18S rRNA*, *28S rRNA*, *12S rRNA*, *cox1*, *histone H3,* and the elongation factor 1α and found that certain families in Caenogastropoda and the branches of Littorinimorpha clustered together. Cunha et al. [21] sequenced seven Neogastropoda species to obtain complete mitogenome sequences and inferred phylogenetic relationships of the Neogastropoda based on 13 PCGs (both at the amino acid and nucleotide levels) from all available caenogastropod mitogenomes, but the ML and BI phylogenetic analyses failed to recover monophyly in Neogastropods. Osca et al. [51] performed phylogenetic inference of Caenogastropoda using the mitogenome and nuclear datasets, respectively, and found that Architaenioglossa is located within Littorinimorpha but Architaenioglossa is not clustered into a single branch, and this branch is intertwined with Littorinimorpha, which in turn is intertwined with Neogastropoda. But on the whole, their distinctions are relatively clear and only a few branches are not clearly delineated from each other. 

Cunha et al. [21] suggested that there are two reasons for these divergences: there is a convergence of morphological features in Neogastropoda and past molecular studies have not provided sufficient molecular information to resolve the complex phylogenetic relationships of Neogastropoda. In molecular studies in recent years, Conoidea has been shown to be a monophyletic population [52,53,54]. We zoomed in on Caenogastropoda and found that Ponder et al. [3] performed a comprehensive phylogenetic analysis of Caenogastropoda using 164 morphological characters (especially external morphological characters) and 55 specimens, all of whose analyses supported the monophyly of Caenogastropoda. Based on multi-gene sequences and a large number of analyzed samples, Zou [55] concluded that Neogastropoda are shown to be a monophyletic group with a large degree of support in the evolutionary trees of all the integrated data. The monophyly of Neogastropoda is comprehensively verified and validity of using morphological characters to study the phylogeny of Neogastropoda is demonstrated. In this study, our phylogenetic analyses using 13 PCGs also confirmed that Neogastropoda is monophyletic, and we found that the three orders are clearly separated from each other and that there is no clustering of branches between Littorinimorpha and Architaenioglossa, which is in support of the morphological study. We believe that a larger amount of data will help us to solve the problem of confusing species classification, and more research is needed to investigate whether Neogastropoda is monophyletic or not.

## 5. Conclusions

The mitogenome length of *L. leucotropis* is 16,362 bp, containing 37 genes, including 13 PCGs, 2 rRNAs, and 22 tRNAs. Compared with species in the same family, its D-loop region is slightly longer, so the total length is relatively longer. The content of AT bases is 69.20%, which is significantly higher than that of CG bases. Comparing the amino acid content, Phe was found to have the highest content of amino acids, while Arg was found to have the lowest content of amino acids. The secondary structure of *L. leucotropis* tRNA conforms to the typical characteristics of clover structure, but a few tRNA have a smaller ring on the amino acid receptor arm. The selection pressure analysis of the Turridae showed that the Ka/Ks values of all PCGs were less than 1, and the evolution of the species was influenced by purification selection. Especially, the Ka/Ks value of the *cox1* gene was only 0.042, which can prove the stability of the *cox1* gene. Therefore, it is often used as a DNA barcode to identify species. Furthermore, by comparing the gene sequences with those of other species in Turridae, it was found that the sequence was consistent with the Conoidea consensus order, which was relatively conservative, and there was a small amount of rearrangement in other species within Turridae. Most importantly, our phylogenetic analysis of Caenogasteroda using 13 PCGs showed clear boundaries between the three orders, each clustered together with related species and exhibiting monophyly. There have been multiple discussions in recent years about whether Neoastropoda is monophyletic, and the results of this analysis can serve as a validation of the evolutionary status of Neoastropoda.

## Figures and Tables

**Figure 1 animals-14-00192-f001:**
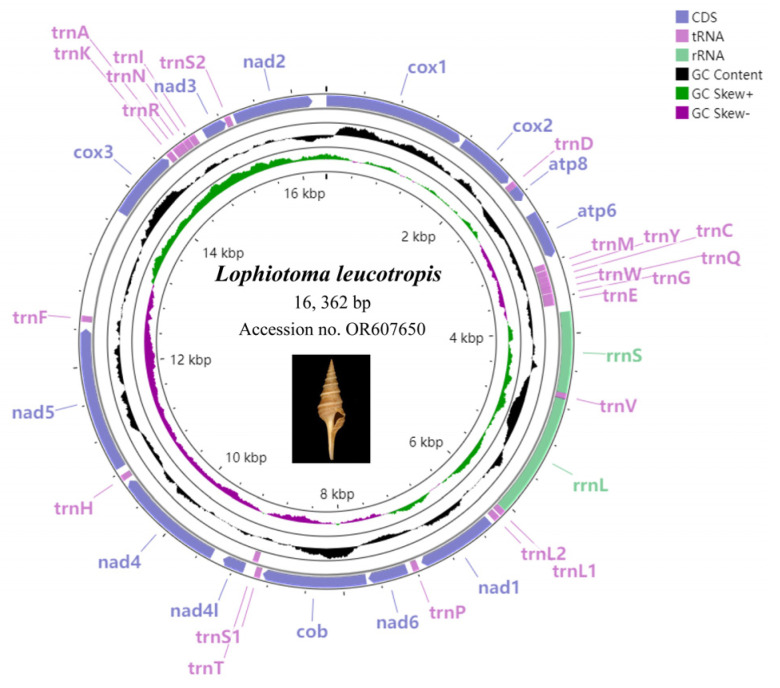
Complete mitogenome circles map of *L. leucotropis*, blue represents PCGs, purple represents tRNA, green represents rRNA, black areas represent GC content; in addition, the map also shows the GC-skew value. Photo by: https://www.marinespecies.org/, accessed on 10 September 2023.

**Figure 2 animals-14-00192-f002:**
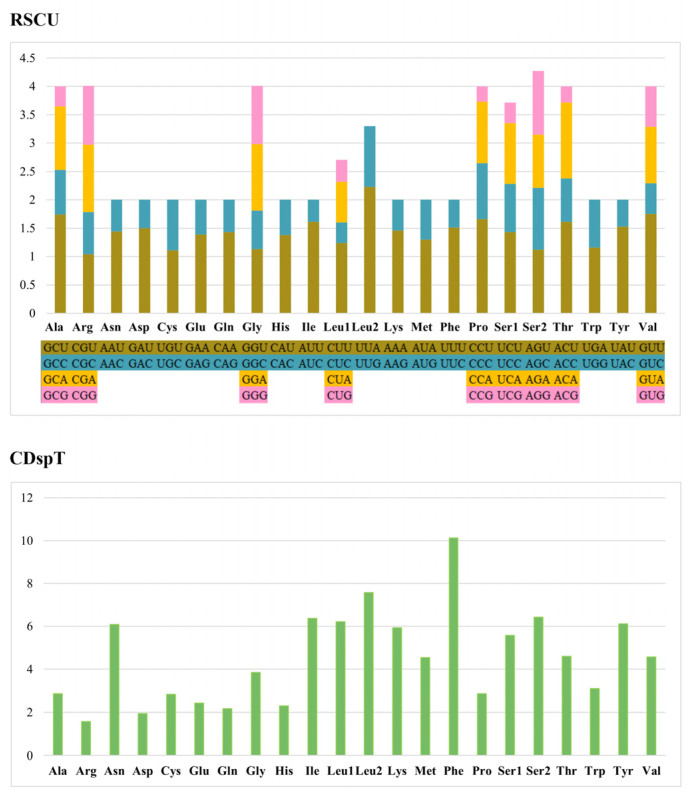
Amino acid content and relative synonymous codon usage (RSCU) of PCGs in *L. leucotropis*.

**Figure 3 animals-14-00192-f003:**
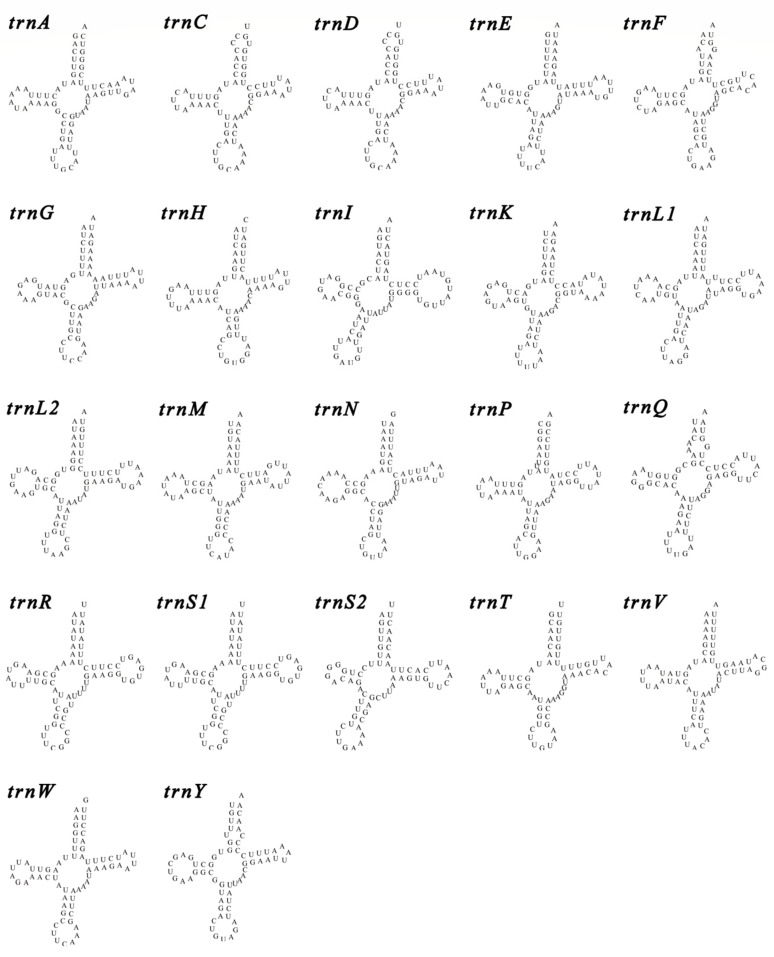
Presentation of the tRNA secondary structure of *L. leucotropis*.

**Figure 4 animals-14-00192-f004:**
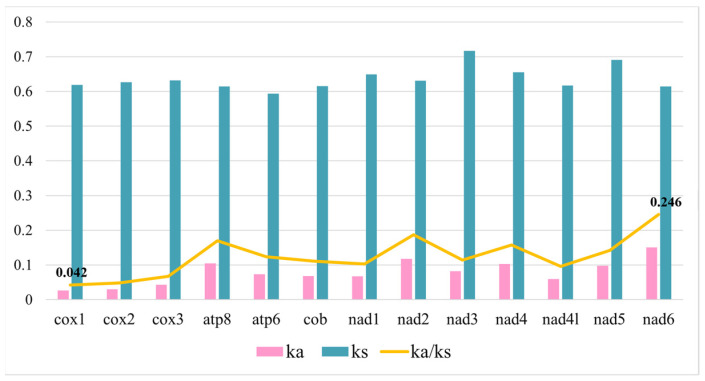
Ka/Ks ratios for all 13 PCGs of 8 Turridae species; pink corresponds to Ka value, blue corresponds to Ks value, and yellow line graph refers to Ka/Ks value.

**Figure 5 animals-14-00192-f005:**
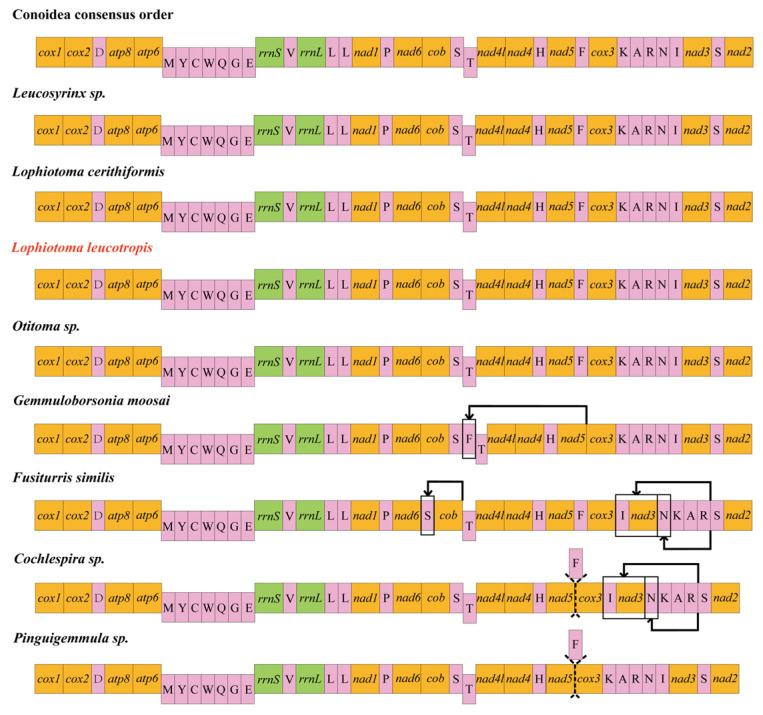
Complete mitogenome alignment map of 8 species in the Turridae; orange for PCGs, pink for tRNA, green for rRNA. The dashed line represents the genes that should have existed here, the arrow indicates the location of gene movement, and the red Latin name represents the species sequenced in this study.

**Figure 6 animals-14-00192-f006:**
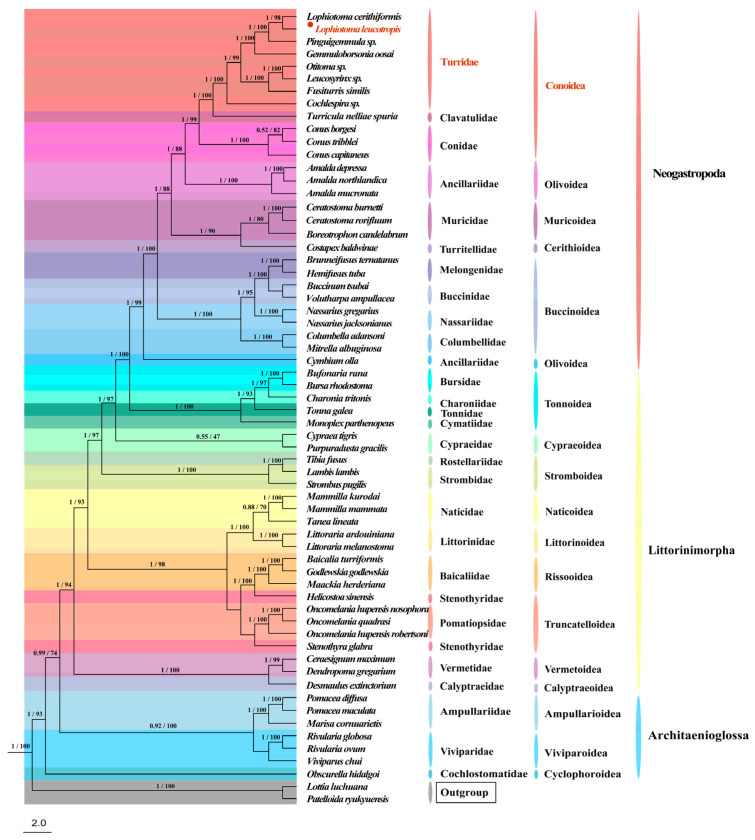
Phylogenetic tree of Caenogastropoda based on 13 PCGs constructed using the BI and ML methods; *L. leucotropis* is marked with black dots and the number on each branch is the Bayesian posterior probability/ML bootstrap support. The red Latin name indicates the sequenced specie in this study, it also indicates the family to which it belongs.

**Table 1 animals-14-00192-t001:** List of Caenogastropoda species for phylogenetic analysis, newly sequenced *L. leucotropis* labeled with •.

Order	Superfamily	Family	Species	Accession No.
Littorinimorpha	Calyptraeoidea	Calyptraeidae	*Desmaulus extinctorium*	NC_079658
	Cypraeoidea	Cypraeidae	*Cypraea tigris*	MK783263
			*Purpuradusta gracilis*	NC_072228
	Littorinoidea	Littorinidae	*Littoraria ardouiniana*	NC_066085
			*Littoraria melanostoma*	NC_064398
	Naticoidea	Naticidae	*Mammilla kurodai*	NC_046596
			*Mammilla mammata*	NC_046597
			*Tanea lineata*	NC_050662
	Rissooidea	Baicaliidae	*Baicalia turriformis*	NC_035869
			*Godlewskia godlewskia*	NC_035870
			*Maackia herderiana*	NC_035871
	Stromboidea	Rostellariidae	*Tibia fusus*	NC_065371
		Strombidae	*Lambis lambis*	NC_071230
			*Strombus pugilis*	NC_059922
	Tonnoidea	Tonnidae	*Tonna galea*	NC_082277
		Cymatiidae	*Monoplex parthenopeus*	NC_013247
		Bursidae	*Bufonaria rana*	NC_054277
			*Bursa rhodostoma*	MW316792
		Charoniidae	*Charonia tritonis*	NC_082220
	Truncatelloidea	Pomatiopsidae	*Oncomelania hupensis nosophora*	LC276226
			*Oncomelania hupensis robertsoni*	LC276228
			*Oncomelania quadrasi*	LC276227
		Stenothyridae	*Stenothyra glabra*	NC_080968
			*Helicostoa sinensis*	OP503947
	Vermetoidea	Vermetidae	*Ceraesignum maximum*	NC_014583
			*Dendropoma gregarium*	NC_014580
Neogastropoda	Buccinoidea	Columbellidae	*Mitrella albuginosa*	NC_066081
			*Columbella adansoni*	KP716637
		Buccinidae	*Buccinum tsubai*	NC_063974
			*Volutharpa ampullacea*	NC_067974
		Melongenidae	*Brunneifusus ternatanus*	NC_059758
			*Hemifusus tuba*	MN462591
		Nassariidae	*Nassarius gregarius*	NC_062791
			*Nassarius jacksonianus*	NC_041548
	Conoidea	Turridae	*Cochlespira sp.*	MH308394
			*Fusiturris similis*	NC_013242
			*Gemmuloborsonia moosai*	NC_038183
			*Leucosyrinx* sp.	NC_038185
			*Lophiotoma cerithiformis*	NC_008098
			• *Lophiotoma leucotropis*	OR607650
			*Otitoma* sp.	MH308405
			*Pinguigemmula* sp.	MH308408
		Conidae	*Conus borgesi*	EU827198
			*Conus capitaneus*	NC_030354
			*Conus tribblei*	NC_027957
		Clavatulidae	*Turricula nelliae spuria*	MK251986
	Muricoidea	Muricidae	*Boreotrophon candelabrum*	NC_046505
			*Ceratostoma burnetti*	NC_046569
			*Ceratostoma rorifluum*	NC_046526
	Cerithioidea	Turritellidae	*Costapex baldwinae*	MW044625
	Olivoidea	Ancillariidae	*Amalda mucronata*	NC_061910
			*Amalda northlandica*	GU196685
			*Amalda depressa*	MN400051
Architaenioglossa	Ampullarioidea	Ampullariidae	*Marisa cornuarietis*	NC_025334
			*Pomacea diffusa*	NC_041142
			*Pomacea maculata*	NC_027503
	Cyclophoroidea	Cochlostomatidae	*Obscurella hidalgoi*	NC_028004
	Viviparoidea	Viviparidae	*Rivularia globosa*	NC_066479
			*Rivularia ovum*	NC_066478
			*Viviparus chui*	NC_035733

**Table 2 animals-14-00192-t002:** Organization of the mitogenome of *L. leucotropis*.

Gene	Position		Strand	Size		Codon		Anticodon	Intergenic Nucleotides
	From	To		Length	Amino Acid	Start	Stop		
*cox1*	1	1545	+	1545	515	ATG	TAA		11
*cox2*	1557	2243	+	687	229	ATG	TAA		−2
*trnD*	2242	2309	+	68				GTC	0
*atp8*	2310	2468	+	159	53	ATG	TAA		2
*atp6*	2471	3166	+	696	232	ATG	TAG		33
*trnM*	3200	3268	-	69				CAT	8
*trnY*	3277	3342	-	66				GTA	1
*trnC*	3344	3407	-	64				GCA	0
*trnW*	3408	3474	-	67				TCA	−2
*trnQ*	3473	3537	-	65				TTG	5
*trnG*	3543	3609	-	67				TCC	−1
*trnE*	3609	3677	-	69				TTC	1
*rrnS*	3764	4646	+	883					−3
*trnV*	4644	4710	+	67				TCA	−10
*rrnL*	4701	6060	+	1360					3
*trnL1*	6064	6132	+	69				TAG	16
*trnL2*	6149	6218	+	70				TAA	0
*nad1*	6219	7160	+	942	314	ATG	TAA		11
*trnP*	7172	7239	+	68				TGG	13
*nad6*	7253	7741	+	489	163	ATT	TAA		3
*cob*	7745	8884	+	1140	380	ATG	TAA		19
*trnS1*	8898	8963	+	66				TGA	0
*trnT*	8964	9029	-	66				TGT	11
*nad4l*	9041	9337	+	297	99	ATG	TAG		−7
*nad4*	9331	10,671	+	1341	447	ATG	TAA		31
*trnH*	10,703	10,768	+	66				GTG	27
*nad5*	10,796	12,484	+	1689	563	ATT	TAA		1
*trnF*	12,486	12,551	+	66				GAA	0
*D-loop*	12,552	13,723	+	1172					0
*cox3*	13,724	14,503	+	780	260	ATG	TAA		12
*trnK*	14,516	14,582	+	67				TTT	19
*trnA*	14,602	14,668	+	67				TGC	−1
*trnR*	14,668	14,737	+	70				TCG	3
*trnN*	14,741	14,809	+	69				GTT	6
*trnI*	14,816	14,886	+	71				GAT	2
*nad3*	14,889	15,242	+	354	118	ATG	TAG		1
*trnS2*	15,244	15,311	+	68				GCT	48
*nad2*	15,360	16,346	+	987	329	ATT	TAA		15

**Table 3 animals-14-00192-t003:** Base content in the mitogenome of *L. leucotropis*.

Gene	A%	C%	G%	T%	A + T%	AT-Skew	GC-Skew
Mito	30.93	15.07	15.73	38.27	69.20	−0.11	0.02
*cox1*	26.98	16.57	18.40	37.98	64.96	−0.17	0.05
*cox2*	29.66	15.90	17.74	36.70	66.36	−0.11	0.05
*atp8*	32.70	10.69	13.84	42.77	75.47	−0.13	0.13
*atp6*	27.55	15.22	15.22	44.51	72.06	−0.24	0.00
*nad1*	27.43	15.63	17.71	39.24	66.67	−0.18	0.06
*nad6*	29.37	13.99	11.89	44.76	74.13	−0.21	−0.08
*cob*	26.86	18.44	15.87	38.83	65.69	−0.18	−0.07
*nad4l*	30.52	14.86	16.06	38.55	69.07	−0.12	0.04
*nad4*	29.86	17.70	13.48	38.96	68.82	−0.13	−0.14
*nad5*	30.27	19.57	12.81	37.35	67.62	−0.10	−0.21
*cox3*	24.74	14.71	21.09	39.45	64.19	−0.23	0.18
*nad3*	24.03	12.40	17.83	45.74	69.77	−0.31	0.18
*nad2*	26.76	12.57	17.19	43.48	70.24	−0.24	0.16

## Data Availability

The mitogenome sequence data of *Lophiotoma leucotropis* were deposited in Genbank with accession numbers OR607650. The data are contained within the article.

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
