# Peer review of "Characterization of *Lophiotoma leucotropis* Mitochondrial Genome of Family Turridae and Phylogenetic Considerations within the Neogastropoda"

_animals, 2024, doi:10.3390/ani14020192_

Round 1

Reviewer 1 Report

Comments and Suggestions for Authors

L29-31 change to ….(A. Adams & ?. Reeve, 1850), which belongs…

L29 name of Reeve also must be included

L31 The shell of L. leucotropis is…

L33 This species often…

L35-38 Taylor in 1980 studied the feeding of predatory gastropods in Hong Kong waters and found that L. leucotropis mainly preyed on a species of Poecilochaetus tropicus in the family Poecilochaetidae, lower selectivity for Laocince and Capitellid [4,5].

L63 it is not clear from this paragraph if Taylor  1993 classification is the most up-to-date?

L68 the explanation what is WoRMS should be included

L73-75 pleas be more precise, except which oceans?

L86 full name of PCG should be written at first mentioning (not only in abstract). 13 PCG of nuclear genome?

L96 I think that the group is monophyletic, paraphyletic or polyphyletic, but not classification

L29-99 I missed a broader and clearer explanation of why Lophiotoma leucotropis was chosen for the mitogenome analysis within Turridae, why this species is so important?  Also, it is not clear how many species of the subclass Caenogastropoda have mitogenomes already been identified and why characterization of one species mitogenome will change dramatically a general understanding of the phylogeny of analyzed group.

L102 single specimen and L103 specimens?

L106 the explanation of WoRMS should be moved in introduction

L109 change “their quality”

L110-112 sorry, but it were no animal experiments, so why it should be written.

L115 what is “QC-qualified”?

L130 so what is GenBank accession number of your sequence?

L136 what do you mean by “annotated by mitos2”?

L161 As authors described the classification of organisms in the group studied sparks debate amoung scientists, so what author classification system was used?

L138- how was alignment of mitogenomes performed?

L165-169 too complicated sentence, please divide into two at least.

L173 what do you mean by “all of which were within the normal range”, what is normal?

L215-220 this is methodological part not results. Here is a commonly known theory. My suggestion to combine 3.3 and 3.4

L246-250 in Methods it was not highlighted that only PCGs sequences but not D-loop, tRNA and rRNA

L262 L must be in italic, genus name

Figure 6, I can't read bootstrap values, these are two small

L271-272 what are the exceptions?

L275-276 so other compared species do not have D-loop? Is this region being not essential for replication of the mtDNA?

L348-351 authors do not used usual criteria, such as conceptualization, methodology, data curation, etc.

L357-359 where were no animal experiments…

Comments on the Quality of English Language

English needs revision, some sentences are difficult to understand

Author Response

Thank you for an opportunity to revise the manuscript. We are also grateful for the constructive comments. We have responded to all the comments point by point below. In addition, all the changes are highlighted in the revised manuscript. We look forward to your approval of the current version or advice on potential additional improvements.

Thank you.

Yingying Ye

Reviewer: 1

  • A point-by-point response to the comments

L29-31 change to ….(A. Adams & ?. Reeve, 1850), which belongs…

Answer:

We double checked and it is indeed (A. Adams & Reeve, 1850);

We have changed it to “which belongs…” (L40).

L29 name of Reeve also must be included

Answer:

This is the record in WoRMs, and the name is indeed like that (40).

L31 The shell of L. leucotropis is…

Answer:

We have replaced the expression according to your suggestion (L42).

L33 This species often…

Answer:

We have replaced the expression according to your suggestion (L44).

L35-38 Taylor in 1980 studied the feeding of predatory gastropods in Hong Kong waters and found that L. leucotropis mainly preyed on a species of Poecilochaetus tropicus in the family Poecilochaetidae, lower selectivity for Laocince and Capitellid [4,5].

Answer:

We added relevant years as suggested and adjusted the position of references (L47-49).

L63 it is not clear from this paragraph if Taylor 1993 classification is the most up-to-date?

Answer:

His classification is not the latest, we have added the relevant content (L79-83).

L68 the explanation what is WoRMS should be included

Answer:

This has been supplemented on L80-81.

L73-75 pleas be more precise, except which oceans?

Answer:

The two references here are quite ancient, and we have searched for other relevant materials, but the wording is quite vague and does not directly indicate in which oceans this species does not exist. So it may not be possible to make changes here.

L86 full name of PCG should be written at first mentioning (not only in abstract). 13 PCG of nuclear genome?

Answer:

The full name of PCGs has been added here (L100);

The 13 PCGs refer to mitogenome, and we have made a supplement to this (L101).

L96 I think that the group is monophyletic, paraphyletic or polyphyletic, but not classification

Answer:

We have removed the inaccurate statements (L111).

L29-99 I missed a broader and clearer explanation of why Lophiotoma leucotropis was chosen for the mitogenome analysis within Turridae, why this species is so important? Also, it is not clear how many species of the subclass Caenogastropoda have mitogenomes already been identified and why characterization of one species mitogenome will change dramatically a general understanding of the phylogeny of analyzed group.

Answer:

There are relatively few known species sequences in Turridae, and this specie itself does not have any uniqueness. Its main function is to enrich data and does not play a direct decisive role.

All the currently sequenced species data of Caenogastroda can be queried in NCBI, which is relatively rich and has reached over a hundred. The 60 data used in this study are only a part of it.

There are three main reasons that affect the phylogenetic analysis results of Neogastropoda: 1) differences in species richness used (quantity or variety). 2) different genes used in the analysis (many molecular studies have used single genes or partial mitochondrial genes, but recent literature suggests that using complete mitochondrial genome data would be more accurate). 3) The methods used in the analysis (BI/ML/NJ...). Therefore, it is not the individual species that affects the overall analysis results. Our goal is to use as much species data as possible on the basis of rich data, and use the complete mitogenome to analyze whether Neogastropoda is monophyletic.

L102 single specimen and L103 specimens?

Answer:

Single specimen, we have corrected it (L117-118).

L106 the explanation of WoRMS should be moved in introduction

Answer:

We have moved it to introduction (L80).

L109 change “their quality”

Answer:

We have changed to “The DNA quality…” (L124).

L110-112 sorry, but it were no animal experiments, so why it should be written.

Answer:

This involves DNA extraction and PCR amplification of marine mollusks, so animal experiments are actually conducted. In our previous submissions, it was suggested that we add this sentence.

However, we still deleted this sentence in 2.1, and we think it would be more appropriate to include it in the final statement of the article.

L115 what is “QC-qualified”?

Answer:

The expression here is inaccurate and has been corrected to “qualified” (L128).

L130 so what is GenBank accession number of your sequence?

Answer:

It is OR607650, in most cases, the GenBank number appears more appropriately in the “Result and Analysis” section, because it belongs to the results we obtained after uploading data. Therefore, we have written it in section 3.1 (L186).

L136 what do you mean by “annotated by mitos2”?

Answer:

The secondary structure of tRNA was annotated by Mitos2. I have reorganized this sentence for better understanding (L151-152).

L161 As authors described the classification of organisms in the group studied sparks debate amoung scientists, so what author classification system was used?

Answer:

The classification of all species used in this table was selected based on the classification in WoRMs, and Worms' classification system is updated in a timely manner, making it more authoritative.

L138- how was alignment of mitogenomes performed?

Answer:

Using the software MEGA-X, compare the COI gene obtained from our sequencing with the COI gene fragments downloaded from NCBI. We have added the relevant content here to section 2.2 (L138-140).

L165-169 too complicated sentence, please divide into two at least.

Answer:

We have broken down the sentences (L185-190).

L173 what do you mean by “all of which were within the normal range”, what is normal?

Answer:

It refers to the fact that the length of these genes is reasonable within Gastropoda. There are significant differences in the length and size of different genes in the mitogenome, such as atp8 being around 150bp and nad5 being around 17000bp. There may be differences among different species, but the fluctuations always remain within a small range.

L215-220 this is methodological part not results. Here is a commonly known theory. My suggestion to combine 3.3 and 3.4

Answer:

We have consulted relevant references and found that the basic feature analysis and selection pressure analysis of genes are written separately, so we think it is more appropriate to keep the two paragraphs.

We have moved some methodological part to section 2.3 (153-156).

L246-250 in Methods it was not highlighted that only PCGs sequences but not D-loop, tRNA and rRNA

Answer:

We have made supplements in the methods (L158).

L262 L must be in italic, genus name

Answer:

We have made corrections here (L281).

Figure 6, I can't read bootstrap values, these are two small

Answer:

The image has been adjusted, the bootstrap values have been enlarged and some species have been removed for clearer visibility.

L271-272 what are the exceptions?

Answer:

Lottiidae, belonging to Patelogastropoda, Gastropoda, has appeared in 38 and 39 mitochondrial genes in this family, mainly tRNA (TrnW or TrnM) that has undergone replication. Because this is not the main topic of discussion, there is no expanded description.

L275-276 so other compared species do not have D-loop? Is this region being not essential for replication of the mtDNA?

Answer:

The other seven species in this family also exist in the D-loop region, but we found that L. Leucotropis has a longer D-loop region, which results in its gene length being longer than several other species.

The expression in the article is not accurate enough, we have made revisions (L293-295).

L348-351 authors do not used usual criteria, such as conceptualization, methodology, data curation, etc.

Answer:

We have made changes according to the submission format (L388-393).

L357-359 where were no animal experiments…

Answer:

Because the samples used in this experiment belong to marine mollusks and involve experimental processes such as such as DNA extraction and PCR amplification. In our team's submissions over the past few years, both reviewers and editors have suggested that we add this sentence to the article. Therefore, we think it is better to keep this sentence here.

Reviewer 2 Report

Comments and Suggestions for Authors

This study is devoted to the analysis of Lophiotoma leucotropis (Turridae) mitochondrial genome and the construction of Neogastropoda phylogeny in order to verify the monophyly of this morphologically very diverse group.

The analysis was carried out at an up-to-date level, I have no complaints about the choice of methods and interpretation of the results obtained.

I have only some minor changes.

1)    Authors should check the places where abbreviations are described - it is not always done at the first mention (for example, the line 68, WoRMS), it is not necessary to describe in detail the decryptions already entered once throughout the text (ex, ML and BI).

2)    Add please author and year for the first mention of species names where missed (line 143, 233-240, etc.)

3)    Line 216-217: the designation of abbreviations is mixed up (Ka - non-synonymous, Ks - synonymous)

4)    Line 263-264: the correct spelling is Bayesian posterior probability / ML bootstrap support.

5)    Maybe it's worth collapsing some genera to improve the visibility of the focus group on Fig.6?

6)    Phrase on the D-loop (lines 275-576) confused me. Do other species not have a D-loop?

These and some more edits are marked in the pdf file attached.

Author Response

Thank you for an opportunity to revise the manuscript. We are also grateful for the constructive comments. We have responded to all the comments point by point below. In addition, all the changes are highlighted in the revised manuscript. We look forward to your approval of the current version or advice on potential additional improvements.

Thank you.

Yingying Ye

Reviewer: 2

This study is devoted to the analysis of Lophiotoma leucotropis (Turridae) mitochondrial genome and the construction of Neogastropoda phylogeny in order to verify the monophyly of this morphologically very diverse group.

The analysis was carried out at an up-to-date level, I have no complaints about the choice of methods and interpretation of the results obtained.

I have only some minor changes.

  • Authors should check the places where abbreviations are described - it is not always done at the first mention (for example, the line 68, WoRMS), it is not necessary to describe in detail the decryptions already entered once throughout the text (ex, ML and BI).

Answer:

We have modified these details, such as L80, L100, L267, L281, L338.

  • Add please author and year for the first mention of species names where missed (line 143, 233-240, etc.)

Answer:

We have added author and year to the first mention of species (L48-49, L162, L163, L253, L254, L255), except for some that have not been identified using “sp.” notation.

  • Line 216-217: the designation of abbreviations is mixed up (Ka - non-synonymous, Ks - synonymous)

Answer:

I made a fundamental mistake and it has been corrected here (L237-238).

  • Line 263-264: the correct spelling is Bayesian posterior probability / ML bootstrap support.

Answer:

We have corrected it (L282).

  • Maybe it's worth collapsing some genera to improve the visibility of the focus group on Fig.6?

Answer:

We have removed 20 species and adjusted the images for clearer visibility.

  • Phrase on the D-loop (lines 275-276) confused me. Do other species not have a D-loop?

Answer:

Other species also exist in the D-loop region, and my expression here is not accurate enough. I have made corrections (L293-295).

These and some more edits are marked in the pdf file attached.

Answer:

Thank you for pointing out the issues. We have made revisions one by one.

Round 2

Reviewer 1 Report

Comments and Suggestions for Authors

11)       Simple Summary is too short. According to requirements it should be no more than 200 words. I suggest, describe at the beginning why the research is important and at the end to highlight future perspectives.

22)       My previous remark “L29-99 I missed a broader and clearer explanation of why Lophiotoma leucotropis was chosen for the mitogenome analysis within Turridae, why this species is so important?  Also, it is not clear how many species of the subclass Caenogastropoda have mitogenomes already been identified and why characterization of one species mitogenome will change dramatically a general understanding of the phylogeny of analyzed group.

Thank you for your reply, but my primary intention of this remark was slightly different from your answers. Please improve the end of introduction to explain why the species was chosen for the study and the importance of phylogenetic studies of the group analysed. My remark does not imply a bad study or a bad study design, but the comment refers to the need for improvement by highlighting and explaining the relevance of the research carried out and why the research object was chosen. Please include 3-5 sentences in the last or second-to-last paragraph of introduction.

33)       L393 I understand the exaggerated ethical claims of the journals, but I don't agree with them in all cases, it is necessary to look at the details of each case. I do not agree in this phrase “all animal experiments” “experiments”. I have also long-term experience when journals apply nonrealistic ethical requirements, when it is no need based on national and international requirements. DNA extraction and PCR is not experiments with animals, its manipulation of animal’s material. You have not done parasite transmission experiments, no behavioral experiments, no physiological/biochemical experiments, etc. My suggestion: “The use of animals for the study was conducted under the guidance approved by the Animal Research and Ethics Committee of Zhejiang Ocean University (NO: 394 ZJOU20180166). Institutional Review Board Statement Not applicable.”

44)       L135-138 please revise English

Comments on the Quality of English Language

   L135-138 please revise English

Author Response

Simple Summary is too short. According to requirements it should be no more than 200 words. I suggest, describe at the beginning why the research is important and at the end to highlight future perspectives.

Answer: We have added relevant content in the “Simple Summary” section (L10-14, L20-23).

1)  My previous remark “L29-99 I missed a broader and clearer explanation of why Lophiotoma leucotropis was chosen for the mitogenome analysis within Turridae, why this species is so important? Also, it is not clear how many species of the subclass Caenogastropoda have mitogenomes already been identified and why characterization of one species mitogenome will change dramatically a general understanding of the phylogeny of analyzed group.”

Thank you for your reply, but my primary intention of this remark was slightly different from your answers. Please improve the end of introduction to explain why the species was chosen for the study and the importance of phylogenetic studies of the group analysed. My remark does not imply a bad study or a bad study design, but the comment refers to the need for improvement by highlighting and explaining the relevance of the research carried out and why the research object was chosen. Please include 3-5 sentences in the last or second-to-last paragraph of introduction.

Answer: I understand what you mean this time. The last paragraph of the Introduction section is not well connected, which means that the emphasis on the topic is not clear enough… Therefore, I have added a few sentences here to explain clearly the reason for using the mitochondrial genome of L. leucotropis for analysis, and constructing the Caenogastropoda phylogenetic tree (L109-122).

2)  L393 I understand the exaggerated ethical claims of the journals, but I don't agree with them in all cases, it is necessary to look at the details of each case. I do not agree in this phrase “all animal experiments” “experiments”. I have also long-term experience when journals apply nonrealistic ethical requirements, when it is no need based on national and international requirements. DNA extraction and PCR is not experiments with animals, its manipulation of animal’s material. You have not done parasite transmission experiments, no behavioral experiments, no physiological/biochemical experiments, etc. My suggestion: “The use of animals for the study was conducted under the guidance approved by the Animal Research and Ethics Committee of Zhejiang Ocean University (NO: 394 ZJOU20180166). Institutional Review Board Statement Not applicable.”

Answer: After careful consideration, we believe that your suggestion is reasonable, so we have removed the content regarding the animal statement at the end of the article.

3)  L135-138 please revise English

Answer: We have reorganized the expression. (L154-157). In addition, we also made some formatting changes to the article and references.